# Cafeteria Online: Nudges for Healthier Food Choices in a University Cafeteria—A Randomized Online Experiment

**DOI:** 10.3390/ijerph182412924

**Published:** 2021-12-08

**Authors:** Christine Kawa, Patrizia M. Ianiro-Dahm, Jan F. H. Nijhuis, Wim H. Gijselaers

**Affiliations:** 1Department of Management Sciences, University of Applied Sciences Bonn-Rhein-Sieg, 53359 Rheinbach, Germany; patrizia.ianirodahm@h-brs.de; 2Department of Educational Research and Development, School of Business and Economics, Maastricht University, 6211 LM Maastricht, The Netherlands; jfh.nijhuis@maastrichtuniversity.nl (J.F.H.N.); w.gijselaers@maastrichtuniversity.nl (W.H.G.)

**Keywords:** food choice, body shape nudge, cafeteria, health intervention, nudge awareness

## Abstract

Many people do not consume as much healthy food as recommended. Nudging has been identified as a promising intervention strategy to increase the consumption of healthy food. The present study analyzed the effects of three body shape nudges (thin, thick, or Giacometti artwork) on food ordering and assessed the mediating role of being aware of the nudge. Students (686) and employees (218) of a German university participated in an online experimental study. After randomization, participants visited a realistic online cafeteria and composed a meal for themselves. Under experimental conditions, participants were exposed to one out of three nudges while choosing dishes: (1) thin body shape, (2) thick body shape, and (3) the Giacometti artwork nudge. The Giacometti nudge resulted in more orders for salad among employees. The thin and thick body shape nudges did not change dish orders. Awareness of the nudge mediated the numbers of calories ordered when using the Giacometti or thin body shape nudges. These findings provide useful insights for health interventions in occupational and public health sectors using nudges. Our study contributes to the research on the Giacometti nudge by showing its effectiveness when participants are aware (it is effective under conditions where it is consciously perceived).

## 1. Introduction

Several national surveys have shown that Germans do not consume enough fruit and vegetables, falling short of the amounts suggested by guidelines for healthy eating [1,2,3]. In 2018, German insurance companies spent EUR 158 million on health promotion and prevention measures, the majority involving nutrition. These measures frequently concern higher education settings [4]. According to the Ottawa Charter approach [5], focusing on promoting health within individuals’ various life settings, the German population needs to improve their eating habits within their occupational settings. This setting is especially promising for health interventions because German employees spend almost 35 h per week at work [6], and the majority of German employees (65.4%; *N* = 2627 employees) visit a cafeteria at lunchtime [7]. Employers, likewise, should adopt more cost-effective nutrition interventions.

One effective and cheap health intervention in food choice and nutrition across the entire life span includes so-called nudges [8,9,10,11]. Nudges change the decision-making environment by promoting a specific choice. They usually help individuals avoid, exploit or harness biases and cognitive errors that may occur in their decision-making [12]. Nudges can be classified according to the cognitive system activated in an individual. Type 1 nudges exert influence via the automatic cognitive system not involving deliberation. Type 2 nudges influence one’s attention and reflective thinking processes exert influence through deliberation [13]. Several meta-analyses and systematic reviews support the positive effects of nudges on improving eating behaviors [14], including in university and cafeteria settings [15,16,17,18]. For example, nudging within a cafeteria’s lunch pre-ordering system led to 51.4% more fruit and 29.7% more vegetable orders than a pre-ordering system without nudging [19]. Nudge interventions that involved applying a poster in a canteen increased fruit and vegetable consumption [20]. Although nudges are effective in cafeteria settings, effect sizes are often small [18]. A meta-analysis of healthy eating nudges involving field experiments revealed a Cohen’s *d* = 0.23 [21]. We propose that universities may be able to promote students’ and employees’ healthy eating by applying healthy eating nudges in a cafeteria setting.

The so-called Giacometti cue exemplifies an effective nudge for decreasing unhealthy food consumption and increasing healthy food consumption [16,22,23,24,25]. This nudge uses the artwork of Alberto Giacometti (concerning a thin, genderless, human-like figure) and would be classified as a type 1 nudge [13]. So far, the Giacometti nudge has produced a medium effect size of Cohen’s *d* = 0.39–0.65 [23]. Despite the effectiveness of the Giacometti nudge’s thin figure, other research on thin body shape cues has shown contrasting [26] or insignificant effects [27,28]. Additionally, being exposed to a thick body shape either increased unhealthy food consumption [29,30] or decreased the choices for unhealthy meals [31]. These inconsistent research results regarding thin or thick human body shape cues call for further research on body shape nudges.

The issue has recently been raised that we still lack understanding of the exact workings of a nudge, as well as the conditions under which an individual is susceptible to nudge effects [32,33]. Additionally, a publication bias creating a positively skewed impression of nudge effectiveness may exist [10]. A recent systematic review identified differences in nudge effectiveness regarding several demographic variables (such as education and occupation). Nudges are not equally effective for different target groups. These effects are called equity effects and have been found for dietary nudges [34]. Assessing nudge effects separately for different target groups reached by a nudge (such as students and employees in a university’s cafeteria) is important in order to identify possible equity effects.

Several studies have assessed whether being aware of a nudge and/or its purpose plays a role in its effectiveness. Research on the Giacometti nudge has also explored the possibility that awareness of its presence may cause reactance and hinder its effectiveness [22,25]. However, research on the issue of nudge awareness is scarce and inconclusive [25,35,36]. Some studies found that disclosing a nudge’s presence did not make a difference regarding its effectiveness [31,37]. Other studies have found that transparent nudges aimed at healthy eating behaviors cause behaviors opposing the nudge’s intent [32]. The degree to which an individual is aware of a nudge may play a crucial role in its effectiveness but has not yet been determined [25].

Considering the findings so far, we expected the exposure to a Giacometti nudge to result in more orders for healthy food and fewer for unhealthy food, compared to a control group (hypothesis 1). Furthermore, we expected the thin body shape nudge to mirror the effects of the thin Giacometti figure, also resulting in fewer orders for unhealthy food and more for healthy food as compared to a control group (hypothesis 2). The thick body shape nudge was expected to result in more orders for unhealthy food and fewer for healthy food as compared to a control group (hypothesis 3). Furthermore, we expected the awareness of the nudge’s presence to mediate the number of calories ordered (hypothesis 4).

The present study had two aims. First, we explored the effect of the Giacometti nudge, as well as thin and thick body shape nudges, on healthy and unhealthy food orders in a workplace setting. We assessed two target groups (students and employees) separately to account for possible equity effects in nudging. Secondly, we observed the role played by nudge awareness regarding calorie orders. We contribute to the existing research on body shape nudges by assessing the effectiveness of three different types of nudges. We also accounted for possible equity effects in nudging and concluded the role of awareness in nudging.

## 2. Materials and Methods

### 2.1. Setting and Context

Data were collected from 30 June until 28 July 2020, using a virtual cafeteria setting. All students and employees of a German University of Applied Sciences were invited via an e-mail to participate in an online experiment assessing the selection of different dishes typically available in the university cafeteria (convenience sampling). This population consists of approximately 9812 students and 734 employees. Students are part of one of the following departments: Computer Science (approximately 24.9%); Electrical Engineering, Mechanical Engineering and Technical Journalism (approximately 19.4%); Natural Sciences (approximately 12.3%); Management Sciences (approximately 33.7%); Social Policy and Social Security Studies (approximately 9.7%). Approximately 45.7% of the employees are research/scientific staff, approximately 32.2% are non-research/scientific staff and approximately 21.7% are professors. All participants were informed about the purpose of the study and personal data security. They actively gave their consent to participate. The present study is part of a larger research project which was approved by the ethical committee of the University of Bonn (sequence number 086/19). This project is in line with the ethical standards of the Declaration of Helsinki. This research project’s general aim is to assess the health and well-being of the university’s students and employees as well as to compose health interventions and to offer programs to improve the health and well-being of these individuals in different areas.

We carefully designed the online food choice scenario to make it as realistic as possible (Figure 1), basing each decision on prevalent research in the area of food choice and nudging. We only included dishes in the food choice scenario that are usually sold in the cafeteria [38,39], and presented these dishes in the form of photographs under which a descriptive name was given. We also included pictures of the cafeteria’s actual food displays and portrayed the dishes on the cafeteria’s actual tableware and trays [40,41]. Participants were asked to imagine having their usual budgets and all dishes were priced as usual. We did not include specific prices in the design, because most studies only did so when they manipulated the price of a dish as an independent variable [38]. If the participants did not want to choose any of the dishes portrayed, they were able to choose an empty tray option. At the end of the food choice scenario, participants were shown their choices. Here, they were able to change their choices if they wished to do so.

### 2.2. Study Design and Procedure

Participants entered the virtual food choice scenario and were asked to compose a meal for themselves as they typically would when visiting the university cafeteria [38,39]. Participants were randomly assigned to one of three experimental conditions or a control condition. In the experimental conditions, the food choice scenario depicted one of three nudges: (1) thin male and female body shape, (2) thick male and female body shape, or (3) a thin, genderless Giacometti nudge (Figure 2).

The nudge was displayed on every page of the online experiment [12]. Congruent with an individual’s manner of visual representation, the nudge was displayed in the left-hand corner of the screen [42]. In the control condition, the food choice scenario did not include a nudge, leaving the left-hand corner of the screen blank. Figure 3 displays the food choice scenario in the thin body shape nudge condition.

Based on prevalent research on food choice and nudging, we assessed different dependent variables: (1) the number of healthy dishes (e.g., salads and fruit salads) ordered, (2) the number of unhealthy dishes (e.g., chocolate pudding) ordered, and (3) the numbers of calories ordered [16,22,23,43]. The latter was calculated by assigning a caloric value to each ingredient of a dish and summing up these calorie counts. The caloric values were based on nutritional information from the German Food Association [44]. The caloric values of one dish are as follows: salad has 54 kcal, fruit salad has 83 kcal, and the chocolate pudding has 188 kcal. Before the experiment, we consulted a certified nutritional expert (member of the Association of Oecotrophologen—VDOE e.V.), who categorized salad and fruit salad as healthy and chocolate pudding as unhealthy based on information of the German Food Association and the World Health Organization.

### 2.3. Questionnaire

After choosing the dishes, participants indicated whether they were students or employees. They stated their degree of hunger and their perceived importance of healthy eating on a 5-point scale (*not hungry at all–very hungry; not important at all–very important*). They reported their height (cm) and weight (kg). Furthermore, we assessed self-control with a 5-point scale (*disagree–agree*) using the German version of the *Brief Self–Control Scale* [45], consisting of 13 items (α = 0.87; Appendix A). Only in the experimental conditions did participants state their level of awareness of the nudge’s presence [14] (*strongly disagree–strongly agree*) and completed the *Intervention-elicited Reactance Scale*, measuring whether the exposure to an intervention caused psychological reactance [46]. This scale consisted of six items (α = 0.87) and used a 7-point scale (*strongly disagree–strongly agree*). To protect data privacy, participants’ genders and ages were not assessed. As compensation, all participants had the option to be included in a lottery for one of two EUR 25 vouchers for a zero-waste store.

### 2.4. Data Analysis

To test hypotheses 1–3, we executed separate fourfold chi-squared tests on the frequencies of ordering a salad, fruit salad (healthy foods), and chocolate pudding (unhealthy food). In each analysis, we compared the differences in frequencies between the subsets of nudge conditions. These tests were performed separately for students and employees. For these analyses, we used a significance level of 0.05 and bootstrapping with 1000 bootstrap samples for the percentile bootstrap confidence intervals (confidence level of 95%).

To test hypothesis 4, we applied a mediation analysis using the PROCESS v 4.0 macro for SPSS (IBM Corp., Armonk, NY, USA), developed by Andrew F. Hayes. We analyzed participants in the three nudge conditions. Participants under the no-nudge control conditions were not exposed to a nudge, and did not rate nudge awareness; therefore, these participants were not included. In the analysis, the nudge condition represented the predictor; awareness of the presence of the nudge was the mediator; and the outcome was the number of calories consumed. When dealing with an independent variable that involves more than two categories, applying this variable as a multi-categorical variable is suggested [47]. We used indicator coding to compare the Giacometti nudge condition to the thin and thick body shape nudge conditions, regarding their influence on the number of calories ordered, considering nudge awareness as a mediator. Here, we used a significance level of 0.05 and bootstrapping with 5000 bootstrap samples for the percentile bootstrap confidence intervals (confidence level of 95%). Missing values were deleted listwise. Data are available via Mendeley Data at https://data.mendeley.com/datasets/c5rbb6v2kr/1 (accessed on 21 April 2021) [48].

## 3. Results

### 3.1. Descriptive Statistics

We recruited a sample of *N* = 904 participants (*n* = 686 students; *n* = 218 employees). For the total sample, self-control, the importance of healthy eating (*M* = 4.07; *SD* = 0.79) and satisfaction with one’s own weight (*M* = 3.48; *SD* = 1.45) could be described as rather high. The participants’ hunger was intermediate (*M* = 2.85; *SD* = 1.28). The body mass index (BMI) lay within a normal range (*M* = 23.30; *SD* = 3.7) and was calculated from the participants’ height and weight. The nudges barely elicited reactance within the participants of the experimental conditions (*M* = 2.13; *SD* = 1.15). These participants were rather aware of the presence of the nudge (*M* = 3.46; *SD* = 1.50). Table 1 provides an overview of the descriptive statistics per nudge condition, separately for students and employees. For both samples, no significant differences between the four conditions existed.

Due to data privacy reasons, we were not permitted to collect data on participants’ gender and age. An overview of the cohorts was derived from the university’s internal demographic statistics (internal unpublished document). The employee cohort (47.2% male, 51.8% female) consisted of the following age categories: 19–29 years (12.2%), 30–39 years (27.9%), 40–49 years (20.8%), 50–59 years (29.4%) and 60–68 years (9.7%). The student cohort consisted of 59.9% males and 40.1% females. The age categories were 17–24 years (71.5%), 25–30 years (22.1%), and older than 31 years (6.3%).

### 3.2. The Effects of the Three Nudges

To measure the effects of the three nudges, we conducted chi-square tests (Table 2). We expected the Giacometti nudge (hypothesis 1) and the thin body shape nudge (hypothesis 2) to result in more orders for salad and/or fruit salad and fewer for pudding compared to the no-nudge condition. The thick body shape nudge was expected to result in fewer orders for salad and/or fruit salad and more for pudding compared to the no-nudge condition (hypothesis 3).

For the students, the fourfold chi-squared tests revealed no significant differences regarding the frequency of ordering salad (*p* = 0.194), fruit salad (*p* = 0.160) or chocolate pudding (*p* = 0.869). Therefore, hypotheses 1–3 were not confirmed.

For the employees, the fourfold chi-squared tests revealed significant differences in the frequency of ordering a salad (χ^2^ (3) = 8.570; *p* = 0.036). Participants in the Giacometti nudge condition ordered 11 more salad dishes than participants in the no-nudge condition, yielding 11.5% more salads ordered considering the total number of salads ordered under all conditions (96 salads). Thus, hypothesis 1 was partially confirmed. No significant differences regarding the frequency of ordering fruit salad (*p* = 0.094) or chocolate pudding (*p* = 0.217) were found. Therefore, hypotheses 2 and 3 were not confirmed.

We were able to partially confirm hypothesis 1 for the employees in the Giacometti nudge condition regarding the number of orders of healthy food. They ordered 11.5% more salads than employees in the no-nudge condition. No significant results were found for the sample of students or the sample of employees regarding fruit salad and chocolate pudding. 

### 3.3. The Mediator Effect of Nudge Awareness

We expected the participants’ awareness of the nudge to mediate the effect of each nudge on the number of calories ordered. To assess the mediating effect of nudge awareness, we conducted a mediation analysis and found the following patterns. Firstly, nudge awareness differed between the three nudge conditions (Table 3). A post hoc analysis revealed that participants in the Giacometti nudge condition were more aware of the nudge than participants in the thin body shape nudge condition (*p* = 0.001). No differences in nudge awareness regarding the thick body shape nudge condition were found (*p_Giacometti_* = 0.127; *p_thin_* = 0.402). Secondly, the three nudge conditions did not differ in terms of the number of calories they ordered (*p* = 0.962). Thirdly, a positive and significant correlation of *r* = 0.104 (*p* = 0.011) existed between nudge awareness and the number of calories ordered. The more aware the participants were of the nudge’s presence, the more calories they ordered. Table 3 summarizes the number of calories consumed, the degree of nudge awareness for the three experimental conditions, and the *F*-values comparing the three conditions.

Mediation analysis revealed the different relationships in the model. Firstly, both the thick and thin body shape nudges were significantly negatively related to nudge awareness, as compared to the Giacometti nudge (Table 4). Participants in the thick and thin body shape nudge conditions were less aware of the nudge than participants in the Giacometti nudge condition. Secondly, relative to the Giacometti nudge, both the thick and thin body shape nudges had no significant direct relationship with the number of calories ordered. Thirdly, nudge awareness had a significant and positive influence on the number of calories ordered. Maintaining constant nudge conditions, those participants who were more aware of the nudge also ordered more calories. Additionally, an indirect relationship relative to the Giacometti nudge existed between the thick and thin body shape nudges on the number of calories ordered via nudge awareness (ß = −8.131, 95% CI [−19.859 to −0.103]; ß = −14.703, 95% CI [−30.205 to −3.276], respectively). This shows that awareness acted as a mediator. In relation to participants in the Giacometti nudge condition, those exposed to the thick and thin body shape nudges were less aware of the nudge’s presence and ordered more calories as a result. The main results of the mediation analysis are summarized in Table 4.

Hypothesis 4 was confirmed: awareness of the nudge indirectly affected the number of calories ordered. Participants exposed to the thin or thick body shape nudges (compared to participants exposed to the Giacometti nudge) were less aware of the nudge, resulting in more calories ordered. Participants exposed to the Giacometti nudge (compared to participants exposed to the thin and thick body shape nudges) were more aware of the nudge, also resulting in more calories ordered. Considering the frequencies of healthy dishes ordered (Table 1), participants in the Giacometti nudge condition ordered more salads and fruit salads than participants in the other experimental conditions.

## 4. Discussion

This study aimed to explore the effect of three body shape nudges (Giacometti nudge, thin and thick body shape nudges) on healthy (salad and fruit salad) and unhealthy food (chocolate pudding) orders in a workplace setting, separately assessing two target groups (students and employees). Furthermore, we also analyzed the mediating effects of the awareness of a nudge’s presence regarding the number of calories ordered.

The Giacometti nudge is an example of an effective technique for increasing healthy food consumption and decreasing unhealthy food consumption and purchasing [16,22,24,25]. For example, it increased healthy snack purchases at the university’s vending machines by 37% compared to a control condition [24]. In line with these results, we found that the Giacometti nudge increased the number of salad orders. Employees ordered 11.5% more salads than participants in the control condition.

Thus far, the Giacometti nudge has mainly been effectively applied to samples consisting of relatively older people, ranging between 35 and 49 years of age [22,24]. When applied to a much younger sample of 17-year-old school students, a thin body shape nudge based on the Giacometti nudge yielded no significant results [27]. Our results regarding the Giacometti nudge for the student sample were, likewise, insignificant. The Giacometti nudge appears to be more effective in older participants. A recent systematic review of nudges on dietary choices found that nudges are not equally effective for every target group [34]. This was in line with a nudge applied in a university cafeteria, decreasing healthy food choices in interns, whereas increasing healthy food choices in older employees [32]. Thus, a nudge found to be effective in a specific target group is not necessarily effective in another target group. Pre-existing preferences have been found to moderate nudge effectiveness, making certain individuals more susceptible than others to nudges [33]. Employees and students may differ in their preferences regarding healthy eating, highlighting the need for further research regarding the susceptibility to nudges [33].

Research on thin and thick body shape primes may yield inconclusive results. In some studies, these primes increased healthy food consumption and decreased unhealthy food consumption, whereas other studies have reported an opposite pattern [26,27,28,29,30,31]. It has been argued that social comparison processes may occur when making dietary choices while in the company of other individuals [49]. The degree to which individuals compare themselves to a body shape prime, i.e., as being either thicker or thinner, blurs the effects of such primes. The present study found no significant nudge effects for the thin or thick body shape nudges—either among the employees or the students. We did, however, find rather large standard deviations for the number of calories ordered by individuals in the thin and thick body shape nudge conditions. This implies that these nudges evoked a rather differentiated reaction in the participants. Our nudges were intended to encourage healthy eating behavior, but it is possible that the participants in the thin and thick body shape nudge conditions did not grasp this intention. It is also possible that the thin and thick body shape nudges were not perceived as particularly thin or thick, and therefore did not elicit the intended effect. Consequently, these two nudges did not have the intended effects.

The role played by the awareness of a nudge on its effectiveness has recently gained attention in research. The results are inconsistent [22,25,31,32,37], and more research on this topic is necessary [25,35,36]. The degree to which an individual is aware of a nudge has been suggested to impact its effectiveness [22,25,32,33]. In line with these studies, we found relative indirect nudge effects on the number of calories ordered via nudge awareness. Greater awareness of the Giacometti nudge resulted in ordering more calories by ordering more healthy dishes. Thus far, the Giacometti nudge has been classified as an automatic and unconscious type 1 nudge [13,16]. We found it to be more effective when people were aware of its presence. Accordingly, we could also classify it as a type 2 nudge, exerting its influence via conscious cognitive processes. Although this finding is interesting, it also raises concerns. The more the participants were aware of this nudge, the more calories they ordered in healthy food. Ordering and later consuming more calories may lead to weight gain and health risks, even if these calories come from healthy food. The Giacometti nudge is therefore particularly interesting for individuals who want to eat more healthily, but neither want nor need to reduce their calorie consumption. In contrast, less awareness of the thin body shape nudge resulted in ordering more calories by ordering fewer healthy calories. This nudge is classified as an automatic and unconscious type 1 nudge [13]. Again, this finding raises concerns because participants ordered more calories and simultaneously selected less healthy food. In general, our results emphasize the diverse role that awareness may play in nudging, highlighted by the research so far [22,25,31,32,37]. They also stress that pre-existing preferences play a role in susceptibility to nudge effects [33]. The Giacometti nudge should therefore be applied to individuals with the described pre-existing preferences.

Our findings provide useful practical implications for practitioners in the public food sector, as well as the food industry. The Giacometti nudge was effective in a virtual food choice setting, particularly for individuals who wanted to eat more healthily without reducing their calorie consumption. Virtual food choice settings already exist in many areas of daily life, for example, in online food delivery services, in restaurants using virtual food ordering via tablets or screens, or in pre-ordering systems in lunch cafeterias. Applying an effective nudge in such virtual food choice settings provides an opportunity to improve the food choices of customers easily and cost-effectively [19]. As an example, cafeterias should set up a virtual lunch ordering system in which individuals can conveniently order their lunch beforehand and pick it up at an arranged time. Portraying the Giacometti nudge saliently when individuals compose their meal online has the potential to improve food choices under conditions where an individual’s pre-existing preferences are to eat healthily without counting calories. These preferences should be assessed by the lunch ordering system before ordering to ensure that the appropriate nudge is portrayed.

It is important to note that this study assessed online food choices, not actual food choices. Even though the food choice scenario was carefully designed, differences may exist between online and actual food choices; the unknown validity of virtual food choice settings needs to be considered [50]. Due to the COVID-19 pandemic, we used an online setting instead of a field experiment setting. This online setting nevertheless created standardized conditions regarding nudges. Future studies should still also assess actual food choices and consumption. For data privacy reasons, we were not permitted to elicit participants’ gender or age. Age differences may explain the equity effects found for students and employees. Both factors are known to influence eating behavior [1], and should therefore be assessed in future studies. We did not manipulate awareness in the present study but assessed the degree to which participants perceived the nudge. Further insight could be gained by including awareness as a separate factor in future studies.

## 5. Conclusions

Our results suggest that the Giacometti nudge has the potential to influence the ordering of healthier salads among university employees, especially when making individuals aware of its presence. However, it should not be applied as a one-size-fits-all nudge across different target groups, because equity effects may exist and pre-existing preferences in a target group can play a role in nudge effectiveness. The Giacometti nudge increased the number of calories ordered by ordering more healthy food when aware of it. Thus, it is particularly interesting for university employees who want to eat more healthily without reducing their calorie consumption. We found it to be effective in a virtual setting, which allowed for an easy and cost-effective application while paying attention to an individual’s specific dietary goals and pre-existing preferences. These findings are particularly useful for practitioners in the public food sector and the food industry in general because they can be applied in any setting involving virtual food choices, such as online lunch ordering systems in cafeterias or online food delivery systems. In this way, practitioners have the opportunity to nudge and individualize healthier eating behaviors.

The results also show that awareness of a nudge plays an important, but ambivalent, role in its effectiveness. The Giacometti nudge was effective when individuals were more aware of its presence. We, therefore, classify the Giacometti nudge as a type 2 nudge [13], contributing to the theoretical background of nudging. The role played by the awareness of a nudge appeared to vary across nudges, because the participants of the thin body shape nudge condition were less aware of the nudge and still ordered more, but fewer healthy calories.

## Figures and Tables

**Figure 1 ijerph-18-12924-f001:**
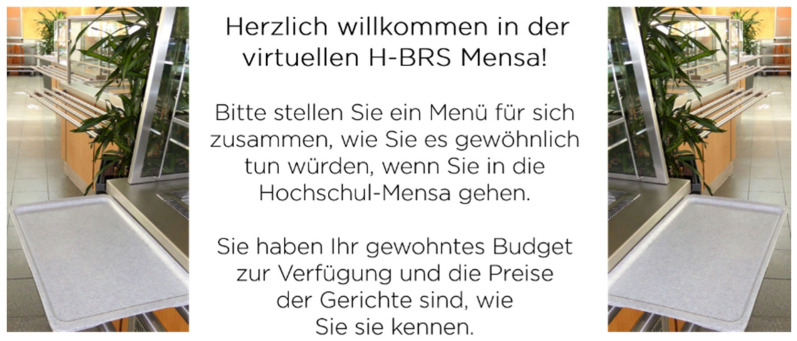
Online food choice scenario depicting the task of composing a typical meal as participants normally would. Participants were asked to imagine having their usual budget as well as the usual prices of the dishes).

**Figure 2 ijerph-18-12924-f002:**
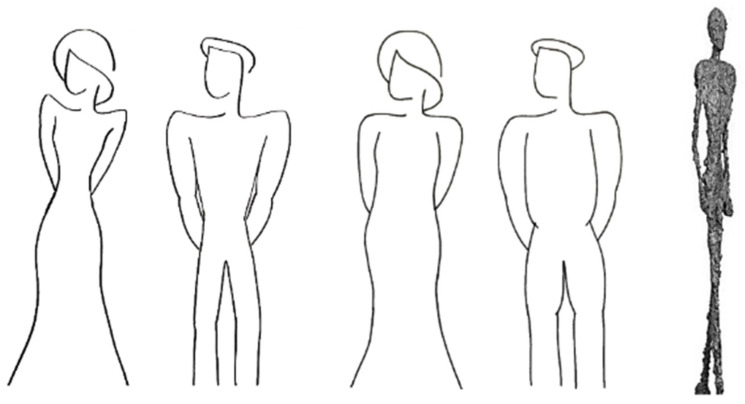
Nudges are shown in the different experimental conditions—from left to right, thin body shape nudge, thick body shape nudge, and Giacometti nudge.

**Figure 3 ijerph-18-12924-f003:**
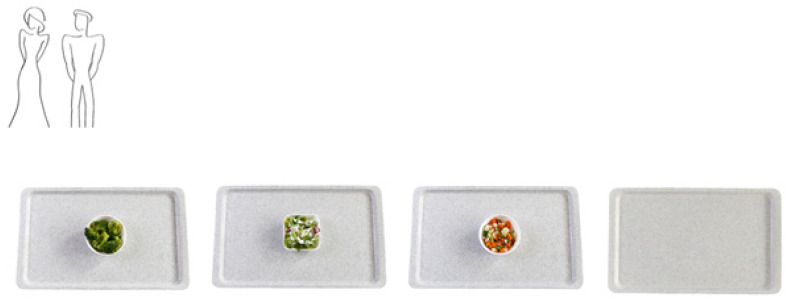
Example of the food choice scenario in the thin body shape nudge condition.

**Table 1 ijerph-18-12924-t001:** Participant characteristics per nudge condition for students and employees.

	**Students (*n* = 686)**			
	**Thin Body Shape Nudge** **(*n* = 167)**	**Thick Body Shape Nudge** **(*n* = 170)**	**Giacometti Nudge** **(*n* = 176)**	**No Nudge** **(*n* = 173)**
BMI	22.97 (3.87)	22.83 (4.48)	22.67 (3.73)	22.62 (3.12)
Hunger ^1^	2.92 (1.19)	2.87 (1.24)	2.83 (1.30)	2.81 (1.28)
Satisfaction with weight ^1^	3.48 (1.25)	3.51 (1.15)	3.54 (1.15)	3.61 (1.01)
Importance of healthy eating ^1^	3.94 (0.91)	3.89 (0.83)	4.05 (0.82)	4.01 (0.78)
Self-control ^1^	3.28 (0.58)	3.26 (0.49)	3.27 (0.54)	3.28 (0.52)
Elicited reactance ^2^	2.15 (2.10)	2.00 (0.90)	2.08 (0.94)	-
Nudge awareness ^1^	3.36 (1.57)	3.65 (1.60)	3.77 (1.43)	-
	**Employees (*n* = 218)**			
	**Thin Body Shape Nudge** **(*n* = 57)**	**Thick Body Shape Nudge** **(*n* = 50)**	**Giacometti Nudge** **(*n* = 64)**	**No Nudge** **(*n* = 47)**
BMI	23.53 (3.19)	23.90 (3.40)	24.33 (3.57)	23.57 (4.20)
Hunger ^1^	2.74 (1.34)	2.76 (1.14)	2.81 (1.37)	3.06 (1.34)
Satisfaction with weight ^1^	3.42 (1.28)	3.40 (1.20)	3.32 (3.23)	3.57 (1.30)
Importance of healthy eating ^1^	4.23 (0.69)	4.14 (0.86)	4.17 (0.73)	4.09 (0.69)
Self-control ^1^	3.51 (0.57)	3.57 (0.42)	3.49 (0.55)	3.49 (0.39)
Elicited reactance ^2^	2.20 (1.12)	2.13 (0.84)	2.20 (0.99)	-
Nudge awareness ^1^	3.03 (1.49)	3.04 (1.67)	3.88 (1.23)	-

^1^ Scale from 1 to 5, with 5 indicating higher values; ^2^ scale from 1 to 7, with 7 indicating higher values; standard deviations in parentheses.

**Table 2 ijerph-18-12924-t002:** Frequencies of the dishes ordered and inferential analyses for the students and employees.

	**Students (*n* = 686)**				
	**Thin Body Shape Nudge (*n* = 167)**	**Thick Body Shape Nudge (*n* = 170)**	**Giacometti Nudge (*n* = 176)**	**No Nudge (*n* = 173)**	**Chi-Squared Tests**
Salad	71 ^a^ (42.5%)	85 ^a^ (50.0%)	83 ^a^ (47.2%)	68 ^a^ (39.3%)	χ^2^ (3) = 4.719; *p* = 0.194
Fruit salad	46 ^a^ (27.5%)	38 ^a^ (22.4%)	57 ^a^ (32.4%)	54 ^a^ (31.2%)	χ^2^ (3) = 5.163; *p* = 0.160
Pudding	45 ^a^ (26.9%)	50 ^a^ (29.4%)	49 ^a^ (27.8%)	44 ^a^ (25.4%)	χ^2^ (3) = 0.717; *p* = 0.869
	**Employees (*n* = 218)**				
	**Thin Body Shape Nudge (*n* = 57)**	**Thick Body Shape Nudge (*n* = 50)**	**Giacometti Nudge (*n* = 64)**	**No Nudge (*n* = 47)**	**Chi-Squared Tests**
Salad	22 (38.6%)	15 (30.0%)	35 (54.7%)	24 (51.1%)	χ^2^ (3) = 8.570; *p* = 0.036
Fruit salad	22 ^a^ (38.6%)	11 ^a^ (22.0%)	28 ^a^ (43.8%)	15 ^a^ (31.9%)	χ^2^ (3) = 6.399; *p* = 0.094
Pudding	9 ^a^ (15.8%)	10 ^a^ (20.0%)	6 ^a^ (9.4%)	11 ^a^ (23.4%)	χ^2^ (3) = 4.447; *p* = 0.217

^a^ subset of nudge conditions that does not differ significantly; percentages indicate the number of dishes chosen in regard to the number of dishes not chosen.

**Table 3 ijerph-18-12924-t003:** Descriptive statistics for the number of calories ordered and nudge awareness were presented per experimental condition, as well as *F*-values comparing these conditions.

	Thin Body Shape Nudge (*n* = 194)	Thick Body Shape Nudge (*n* = 188)	Giacometti Nudge (*n* = 209)	*F*
Calories ordered	862.56 (366.47)	873.31 (403.22)	869.57 (389.06)	*F* (2, 588) = 0.038 *p* = 0.962
Nudge awareness ^1^	3.32 (1.56)	3.56 (1.58)	3.87 (1.37)	*F* (2, 590) = 6.51 *p* = 0.002

^1^ Scale from 1 to 5, with 5 indicating higher awareness; standard deviations in parentheses.

**Table 4 ijerph-18-12924-t004:** Main results of the mediation analysis.

	Nudge Awareness	Number of Calories Ordered		
		Direct	Indirect	Total
	Coefficient (*SE*)	Coefficient (*SE*)	Coefficient (*SE*)	Coefficient (*SE*)
Constant	3.866 (0.104) ***	765.545 (48.757) ***		869.569 (26.730)
X1 (Giacometti vs. thick)	−0.302 (0.151) *	11.871 (38.796)	−8.131 (5.137)	3.739 (38.844)
X2 (Giacometti vs. thin)	−0.546 (0.150) ***	7.696 (38.780)	−14.703 (6.774)	−7.008 (38.526)
Nudge awareness ^1^		26.907 (10.568) *		

^1^ Scale from 1 to 5, with 5 indicating higher values; * *p* < 0.05, *** *p* < 0.001.

## Data Availability

The data presented in this study are openly available on Mendeley Data at https://data.mendeley.com/datasets/c5rbb6v2kr/1.

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
