# Peer review of "Cafeteria Online: Nudges for Healthier Food Choices in a University Cafeteria—A Randomized Online Experiment"

_ijerph, 2021, doi:10.3390/ijerph182412924_

Round 1
Reviewer 1 Report
The present study compared the effectiveness of the Giacometti nudge in ordering healthy and unhealthy foods with that of the thin and thick body shape nudge. Additionally, the authors analyzed the association between the awareness of the nudge and the number of calories in the foods that were ordered. However, this manuscript does not meet the quality standards and is not suitable for publication in the International Journal of Environmental Research and Public Health.
How did the author verify that there were no significant differences between online food choices and actual food choices?
There was no information pertaining to age and sex. While the age range of the student group can be estimated, it is difficult to estimate the age range of the employee group. Age and sex are important parameters in the analysis of food choice and calorie consumption. Thus, the analysis undertaken in this study is inadequate.
Why did the authors analyze only the consumption of salad, fruit salad, and pudding? The participants ordered from a regular budget and chose the dishes from all foods. Do the authors have data regarding protein and carbohydrates consumed? Assessment of nutrient intake is necessary to differentiate ordering of healthy foods from ordering of unhealthy foods.
Methods:
Lines 106-107: The author should cite their previous study.
Results:
Line 197: Did you mean to write Table 2 instead of Table 1?
Line 203: When the author performed the chi-square test to compare the differences in the frequency of ordering salad, fruit salad, and pudding between the four groups (thin body nudge, thick body nudge, Giacometti nudge, and no nudge), did the author perform comparisons between all combinations of groups?
Reviewer 2 Report
The manuscript is interesting, well written and clearly presented. The only limitations are as follows:
I suggest to report in an appendix the 13 items of the German version of the Brief Self-Control Scale.
Conclusions are poor as regards practical implications for promoting healthy choices by the use of nudges.
Reviewer 3 Report
The purpose of this article was to assess different food purchasing behavior using a visual nudge. This was an interesting paper and was well written. I appreciated the attentiveness to clarity; however, it resulted in some redundancy of information in the methods and results. I see that as an optional review/editing.
My only comment is in the statement of Hypothesis 3, page 2, line 88. Please clarify. You state 'expected to result in the opposite behavior.." If you could please specify what opposite means.
On a side note, I did not see the 'thick' nudge to be excessively thick, especially without a visual comparison. I am curious about how those figures are perceived -heavy? Or normal - especially given the vast shift in average weight.
Round 2
Reviewer 1 Report
The author responded appropriately to most of my comments. However, some problems remain. My suggestion is: Lines 107-108: If the author cannot cite a previous study, you must provide the details of the study profile. How to choose participants, percentage of all students, all employees, etc.
Author Response
"Please see the attachment."
